# Loss of *Wnt16* Leads to Skeletal Deformities and Downregulation of Bone Developmental Pathway in Zebrafish

**DOI:** 10.3390/ijms22136673

**Published:** 2021-06-22

**Authors:** Xiaochao Qu, Mei Liao, Weiwei Liu, Yisheng Cai, Qiaorong Yi, Jianmei Long, Lijun Tan, Yun Deng, Hongwen Deng, Xiangding Chen

**Affiliations:** 1Laboratory of Molecular and Statistical Genetics, College of Life Sciences, Hunan Normal University, Changsha 430100, China; liaomei325@126.com (M.L.); lww280@126.com (W.L.); cystao@163.com (Y.C.); yqr13307383916@163.com (Q.Y.); longjianmei@126.com (J.L.); lijuntan2005@aliyun.com (L.T.); hdeng2@tulane.edu (H.D.); 2Hunan Provincial Key Laboratory of Animal Intestinal Function and Regulation, College of Life Sciences, Hunan Normal University, Changsha 430100, China; 3Laboratory of Zebrafish Genetics, College of Life Sciences, Hunan Normal University, Changsha 430100, China; dengyun@hunnu.edu.cn

**Keywords:** CRISPR-Cas9, *wnt16*, RNA sequencing, zebrafish, skeletal development

## Abstract

Wingless-type MMTV integration site family, member 16 (*wnt16*), is a wnt ligand that participates in the regulation of vertebrate skeletal development. Studies have shown that *wnt16* can regulate bone metabolism, but its molecular mechanism remains largely undefined. We obtained the *wnt16^−/−^* zebrafish model using the CRISPR-Cas9-mediated gene knockout screen with 11 bp deletion in *wnt16*, which led to the premature termination of amino acid translation and significantly reduced *wnt16* expression, thus obtaining the *wnt16^−/−^* zebrafish model. The expression of *wnt16* in bone-related parts was detected via in situ hybridization. The head, spine, and tail exhibited significant deformities, and the bone mineral density and trabecular bone decreased in *wnt16^−/−^* using light microscopy and micro-CT analysis. RNA sequencing was performed to explore the differentially expressed genes (DEGs). Gene ontology (GO) and Kyoto Encyclopedia of Genes and Genomes (KEGG) analysis found that the down-regulated DEGs are mainly concentrated in mTOR, FoxO, and VEGF pathways. Protein–protein interaction (PPI) network analysis was performed with the detected DEGs. Eight down-regulated DEGs including akt1, bnip4, ptena, vegfaa, twsg1b, prkab1a, prkab1b, and pla2g4f.2 were validated by qRT-PCR and the results were consistent with the RNA-seq data. Overall, our work provides key insights into the influence of *wnt16* gene on skeletal development.

## 1. Introduction

Osteoporosis is a skeletal disease that affects millions of people around the world, particularly postmenopausal women and older men [1]. According to surveys, it is estimated that 50% of women over the age of 50 have low bone quality, and about 25% of women over the age of 60 suffer from osteoporosis disease [2]. It is mainly low bone mass and the degradation of bone tissue microstructure that result in increased bone fragility and increased bone risk [3,4]. The role of the Wnt signaling pathway in bone biology has gained increased attention in recent years. Wnt signaling plays an important role in maintaining bone mass and bone metabolism, and causes bone diseases including osteoporosis and osteoarthritis [5,6].

As a ligand of Wnt signaling pathway, *wnt16* is related to bone mineral density, cortical thickness, bone strength, and fracture risk, and this molecule might be an attractive target for pharmacologic intervention in treating osteoporosis [7,8,9,10]. Previous studies have shown that Sclerostin, WIF1, and SFRPs protein bind to WNT protein to inhibit Wnt/β-catenin signal pathway transduction, thereby affecting the differentiation of osteoblasts, which have become important targets for clinical drug design [11,12,13]. Some studies have shown that *wnt16^−/−^* mice can cause spontaneous fractures when the cortical thickness is low and the cortex porosity is high, but the trabecular bone volume of the mice does not change [14]. Loss of endogenous *wnt16* leads to the loss of cortical bone, while overexpression of *wnt16* induces an increase in trabecular bone mass. *Wnt16*-targeted therapy may help treat postmenopausal trabecular bone loss [14]. Similarly, transgenic mice overexpressing human *wnt16* in osteocytes have been created for identifying the cell-specific role of *wnt16* in bone homeostasis. The results indicate that *wnt16* plays a vital role in maintaining the quality and strength of cortical bone and trabecular bone [15]. *Wnt16* can indirectly inhibit human and mouse osteoclast production by directly acting on osteoclast progenitor cells or by increasing the expression of osteoprotegerin in osteoblasts, and regulate mechanical strain-induced periosteum bone formation through classical Wnt signaling [16].

The researchers behind the *wnt16* gene study are already at work using transcriptome sequencing. A screen was conducted on the Hedgehog signaling pathway and *wnt16* candidate differential gene using RNA-Seq data of Rugao yellow chickens [17]. The Wnt signaling pathway is rich in the Hedgehog signaling pathway, and the Hedgehog signaling pathway has been shown to promote chondrocyte proliferation [18]. Gingiva-derived mesenchymal stem cells (GMSCs) can inhibit the formation and activity of osteoclasts to reduce autoimmune arthritis. RNA-Seq data analysis of CD39 produced by GMSC has shown that GMSC can exert its osteogenic ability through the Wnt/β-catenin pathway, which provides a new potential therapeutic target for osteoporosis [19]. Transcriptome sequencing was performed on the study of mouse bone cells. The results have shown that *wnt16* is a key molecular regulator of osteogenicity, and the Wnt signaling pathway and TGF-β/BMP signaling pathway regulated by *wnt16* in bone have been discovered; further, it also regulates a large number of genes with known bone phenotypes. The non-canonical Wnt/JNK pathway activation transcription factors Fosl 2 and Fosl 1 are the most significant transcription factors in *wnt16*-activated genes, and Mef2c is a potential participant in *wnt16*-mediated signal transduction [20].

At present, the treatment of osteoporosis can reduce the risk of vertebral fracture by as much as 70% by affecting trabecular bone and estrogen, while the risk of non-vertebral fracture can only be reduced by 20% [1,21,22]. Therefore, the researchers seek more ways to study how to better enhance the therapeutic effect. In order to explore the molecular mechanism of the *wnt16* as a potential target for the treatment of osteoporosis, we established a zebrafish *wnt16* gene knockout model, and investigated the change in phenotype of skeletal development. We found that the skeletal structure of the *wnt16* knockout zebrafish had a distinct deformation and bone mineral density, and the number of bone trabecula decreased. RNA-Seq data analysis showed that knocking out the *wnt16* induced altered changes of a series of related genes, and the down-regulated genes were mainly concentrated in mTOR, FoxO, and VEGF pathways.

## 2. Results

### 2.1. Generation of Wnt16^−/−^ Zebrafish

To understand the function of the *wnt16* gene, the CRISPR/Cas9 gene editing technique was used to obtain the knockout mutant of zebrafish *wnt16*. In this process, two target sites were designed in exon 3 of the *wnt16*; the target sites are as follows: a: GACACAAGCCTGTTGGGCAGCGG, b: GGCCTCCTCACCACGGGTCGAGG (Figure 1A). After using linearized P42250 as a template to synthesize specific guide RNA (gRNA), *wnt16* mutant zebrafish was created by co-injection of Cas9 mRNA and gRNA into zebrafish embryos (one-cell stage). DNA sequencing of target-specific PCR products confirmed that the *wnt16* targeted allele carried a deletion of eleven base pairs (bp) in F1 heterozygote (Figure 1B). Compared with wildtype (WT) larvae, *wnt16^−/−^* larvae showed a 5 bp and 6 bp deletion (sites of TCGAG and GCTGCC, red frames in Figure 1C) using PCR assay. As the CRISPR/Cas9 system has potential off-target effects, we analyzed *wnt16^−/−^* sequencing results through PCR assay, Sanger sequencing, and TA clone. No detectable off-target effect was found in F2 *wnt16^−/−^* mutants.

To further verify the reliability of zebrafish *wnt16* gene knockout, we performed protein structure prediction and qRT-PCR (Figure 2). The deletion of 11 bases resulted in frameshift mutations. We used PSIPRED, a highly accurate secondary structure prediction method, to predict the secondary structure [23]. Compared with WT zebrafish, which included 356 amino acids, 17 alpha helix, and 19 beta strand, translation of *wnt16^−/−^* zebrafish terminates prematurely at 106 amino acids, so that it only has 6 alpha helix and 6 beta strands (Figure 2A,B), resulting in truncated protein. Using SWISS-MODLE software to predict the protein tertiary structure [24], it was found that the tertiary structure of *wnt16^−/−^* zebrafish protein was modified, as shown in Figure 2C,D. The qRT-PCR results showed that the transcription level of *wnt16* was significantly lower than that of WT zebrafish (Figure 2E). Taken together, these results suggested that a *wnt16^−/−^* mutant line was successfully generated with the CRISPR/Cas9 system.

### 2.2. Wnt16^−/−^ Zebrafish Identified Significant Differences Based on Skeletal Phenotype Analysis

To evaluate the direct association between *wnt16* gene and skeletal development, we first conducted a whole-mount in situ hybridization (WISH) experiment. We evaluated the expression of *wnt16* in WT and found that *wnt16* was selectively expressed in bone tissues, including the head, pectoral fin bud, and pharyngeal arch cartilage (Figure 3A). To clarify the accurate expression location and change of *wnt16* in zebrafish, the front-view and side-view images at 48 hpf are shown in Figure 3A. Subsequently, we evaluated the gross appearance of the *wnt16^−/−^* mutant zebrafish and WT by macroscopic observation. It could be seen that the mandible of the *wnt16^−/−^* mutant was longer than that of WT and deformities appeared, such as no caudal fin and signs of scoliosis (Figure 3B). Micro-CT images of *wnt16^−/−^* and WT zebrafish revealed substantial differences between these two groups regarding their skeletal morphology (Figure 3C). From the imaging point of view, the mandible of the *wnt16^−/−^* mutant was significantly longer, its spine was slightly curved, and the caudal fin disappeared. Then, we performed bone correlation analysis. Bone structural parameters including bone mineral density (BMD) and trabecular bone number (Tb.N) were quantified using 3Dmed software. It was found that BMD was significantly decreased in *wnt16^−/−^* zebrafish (Figure 3D). Tb.N, a direct three-dimensional parameter, was calculated at the cone of zebrafish. It decreased in the *wnt16^−/−^* zebrafish compared with the WT group (Figure 3E). It is concluded that knockout of *wnt16* has significantly affected the skeletal development of zebrafish.

### 2.3. Transcriptome Sequencing and Differential Expression Gene Profiles

To explore the molecular mechanism of how the *wnt16* gene affects skeletal development and homeostasis, we undertook RNA sequencing analysis. Changes in gene expression were evaluated in WT and *wnt16^−/−^* mutant zebrafish embryos (including yolk-cell, 1-cell, and 4-cell stages), and RNA-Seq was performed on three replicate samples of each cycle. Approximately 1735 million raw paired-end reads were acquired and 1694 million clean reads remained after removing adaptor sequences and discarding low quality reads. The total length of the reads was about 119.27 gigabases (Gb). Around 92% of the clean reads had quality scores over the Q30 value. Over 81% of the total reads were uniquely mapped to the reference genome, with 91% of total mapping (Appendix A).

Relative to the WT zebrafish, the Volcano plots were constructed by integrating both the *p*-value and fold change of each transcript (*p*-adj < 0.05 and |log2 (fold change)| > 1) to show the general scattering of the transcripts and to filter the differentially expressed genes (DEGs) for the *wnt16^−/−^* zebrafish (red and green dots for up-regulated and down-regulated DEGs, respectively; Figure 4A, yolk-cell; Appendix A, 1-cell, 4-cell). Moreover, a hierarchical clustering of DEGs was conducted (Figure 4B, yolk-cell; Appendix A, 1-cell, 4-cell). In order to gain the co-expressed differential genes in the three cell stages and classify the genes expressed in WT and *wnt16^−/−^* mutants, we generated a Venn diagram and obtained a total of 773 co-expressed differential genes (Figure 4C).

### 2.4. The Skeletal Developmental Pathways Are Down-Regulated Following Wnt16 Knockout

To further assess the potential mechanistic functions of 773 co-expressed differential genes, we conducted GO enrichment analysis and KEGG pathway classifications for the up-regulated genes and down-regulated genes through DAVID bioinformatics resources with a *p*-value of 0.05. The analysis was performed to identify GO enrichment in the categories of cellular components, biological processes, and molecular functions. For GO enrichment, the down-regulated genes were significantly enriched in blood vessel development and vasculogenesis, which might be related to skeletal development (Figure 5A). The KEGG pathway found that differential genes were mainly enriched in the mTOR signaling pathway, FoxO signaling pathway, and VEGF signaling pathway (Figure 5B, Table 1). Studies have shown that these signaling pathways interacted with Wnt signaling pathway and could regulate skeletal development [25,26]. The GO and KEGG enrichment analysis of co-expressed up-regulated genes were significantly enriched in mitochondrion, NADH dehydrogenase (ubiquinone) activit, respiratory chain, oxidative phosphorylation, and metabolic pathways (Appendix A). We found that there is no obvious enrichment pathway related to skeletal development. This indicated that knocking out the zebrafish *wnt16* gene may inhibit the expression of bone-related signaling pathways, resulting in the disruption of the normal development of zebrafish skeleton.

To determine the interaction between down-regulated genes related to skeletal development gathered in GO enrichment and KEGG signaling pathways and *wnt16*, we identified a potential PPI network for these DEGs (Figure 5C). The PPI network integrated these down-regulated genes using STRING analysis; there were 15 nodes and 19 edges involved in the establishment of the gene regulation network. The average node degree of 2.53, local clustering coefficient of 0.619, and PPI enrichment *p*-value of 3.59 × 10^−6^ indicated statistically significant PPI enrichment (*p* < 0.001). From the PPI network, we found that *wnt16* forms a network with other genes by forming a connection with the two genes sulf1 and vegfaa. We speculated that the possible mechanism of abnormal zebrafish skeletal development was due to the knockout of *wnt16*, which led to the accumulation of some down-regulated genes in the mTOR signaling pathway, FoxO signaling pathway, and VEGF signaling pathway. These signaling pathways, together with Wnt signaling pathways, are involved in the regulation of developmental processes, thus inhibiting zebrafish skeletal development.

### 2.5. Validation of Selected DEGs Using Quantitative Real-Time PCR (qRT-PCR)

To validate the expression pattern of DEGs identified by RNA-Seq, qRT-PCR analysis was performed and, based on the GO and KEGG pathway enrichment analysis results, eight down-regulated genes were selected, including akt1, bnip4, ptena, vegfaa, twsg1b, prkab1a, prkab1b, and pla2g4f.2 (Table 1). In general, gene transcription levels from the qRT-PCR assays showed a comparable trend and magnitude of changes relative to the transcriptomic results. Based on the qRT-PCR results, we found that the expression of eight genes in the *wnt16^−/−^* group was significantly lower than that in the WT group (Figure 6). Among these genes, ptena and prkab1a were slightly reduced in the yolk-cell period and showed a significant down-regulation in the 1-cell and 4-cell periods, while bnip4, twsg1b, and pla2g4f.2 showed a consistent trend in the three periods, which were all significantly down-regulated. The qRT-PCR results were found to be consistent with RNA-Seq results, thereby confirming the reliability of RNA-Seq data.

## 3. Discussion

In this study, we used the CRISPR/Cas9 system to knock out the *wnt16* gene of zebrafish at a specific target site, then we genotyped it in F1 and found that the mutation was an 11 bp deletion. The ratio of F2 offspring was roughly consistent with Mendelian genetics. Several methods, including the surveyor assay, T7E1 assay, and Sanger sequencing, were used to analyze the off-target effect [27,28]. In protein structure prediction, *wnt16^−/−^* zebrafish encountered a stop codon during translation, resulting in a cut-off protein containing only 106 amino acids. At the same time, the results of qRT-PCR showed that, in *wnt16^−/−^* zebrafish, mRNA expression significantly decreased. In this result, *wnt16^−/−^* is barely expressed because the forward primer was located on the deletion region. *Wnt16^−/−^* mutant line was successfully established.

Furthermore, micro-CT images showed lengthening of the mandible and scoliosis of the *wnt16^−/−^* zebrafish. The results of the bone correlation analysis showed that BMD and Tb.N decreased. Studies have shown that osteoblasts specifically showed conditional knockout of *wnt16* gene in mice, resulting in decreased BMD and bone strength, and increased porosity [14]. Similarly, a robust trabecular bone phenotype in the osteoblast-specific TG *wnt16* mice was found in both males and females and, compared with WT mice, the trabecular bone volume of distal femur in TG mice increased by 3-fold (male) and 14-fold (female) at 12 weeks of age [15].

Subsequently, in the RNA-Seq data analysis, GO enrichment of down-regulated DEGs showed that most genes were mainly concentrated in blood vessel development and vasculogenesis. KEGG signaling pathway analysis results showed that DEGs were mainly focused on metabolic pathways, mTOR, FoxO, and VEGF signaling pathway. Studies have shown that mTOR signaling pathway, FoxO signaling pathway, and VEGF signaling pathway were related to skeletal development [25,26].

Recent studies have shown that mTORC1 is involved in many aspects of the regulation of cartilage development. mTOR can significantly inhibit the formation of cartilage nodules in limb bud cells without affecting the aggregation of precartilage mesenchyme. At the same time, inhibiting the expression of Sox9 can significantly reduce the accumulation of proteoglycan and the expression of chondrocyte markers in ATDC5 [29,30,31]. mTORC1 promotes glutamine-mediated Wnt catabolism and integrated stress response (ISR), thereby inducing the expression of protein anabolism genes necessary for osteoblast differentiation [32,33]. Bone anabolism Wnt ligands (such as wnt3a, wnt7b, or wnt10b) activate mTORC2 via Lrp5 signaling and reprogram glucose metabolism [34]. Previous studies have shown that the role of FoxO in chondrocytes is essential for the normal development of bone, and the role of FoxO in osteoclasts or osteoblast greatly affects bone resorption and formation, thereby affecting bone mass. Elena Ambrogini’s study showed that conditional deletion of FoxO1, 3, and 4 in three month-old mice resulted in the levels of expression of FoxO1, 3, and 4 being decreased by 60–75% in calvaria and vertebrae. The femoral bone mineral densities (BMDs) are decreased by dual energy X-ray absorptiometry (DEXA) and the femoral cortical widths are decreased by compartment-specific analysis by micro-CT [35,36]. FoxOs can also inhibit Wnt signaling and bone formation in osteoblast progenitor cells [37]. In addition, FoxOs reduces bone resorption through direct antioxidant effects on osteoclasts and up-regulation of the anti-osteoclast factor OPG in osteoblast cell lines [37,38]. For the VEGF signaling pathway, studies have shown that vascular endothelial growth factor (VEGF) is an endothelial cell survival factor, which is necessary for the effective coupling of angiogenesis and osteogenesis [39]. Studies have shown that mechanical signals can regulate the alternative splicing of osteoblasts (rat osteosarcoma cell lines and human primary cells) and VEGF-A in vivo, and VEGF is essential for exercise-induced bone hyperplasia [40].

Eight genes verified by qRT-PCR were classified into the following categories: (1) mTOR signaling pathway: akt1 and ptena; (2) FoxO signaling pathway: prkab1a, akt1, prkab1b, ptena, and bnip4; (3) VEGF signaling pathway: akt1, vegfaa, and pla2g4f.2. Protein–protein interaction analysis revealed the relationship between *wnt16* and the genes of the down-regulated signaling pathway. *Wnt16* interacts with down-regulated genes through the Wnt signaling pathway, which leads to the inhibition of the related signaling pathway. Studies have shown that these genes are related to skeletal development, and OPG induces autophagy through the AKT/mTOR/ULK1 signaling pathway, thus inhibiting bone resorption of osteoclast [41]. Akt1 is a key regulator of osteoblasts and osteoclasts [42]. FoxO actions in chondrocytes are critical for normal skeletal development, and FoxO actions in cells of the osteoclast or osteoblast lineage greatly influence bone resorption and formation and, consequently, bone mass. FoxOs also acts in osteoblast progenitors to inhibit Wnt signaling and bone formation [43]. VEGF is required for effective coupling of angiogenesis and osteogenesis, and can enhance fracture healing [44,45].

Wnt signal is considered to be an important factor regulating bone homeostasis. Multiple GWAS studies have shown that genetic variants of *Wnt16* are associated with BMD and fracture risk [7,10,46]. Research by Imranul Alam et al. showed that human *Wnt16* overexpression in osteocytes influences trabecular and cortical bone mass, structure, and strength in mice; *Wnt16* affects the quality and strength of cortical bone and trabecular bone; and this molecule can be used as a therapeutic intervention to treat osteoporosis or other low bone mass and high bone fragility disorders [47]. For the VEGF signaling pathway, inhibition of VEGF signaling pathway and angiogenesis has become a promising method in preclinical research in recent years. Treatment targeting VEGF signaling in OA can be further enhanced by prolonging the half-life of the drug at the joint and by combining it with anabolic growth factors [48]. Tumor cells and osteogenic cells form heterotypic adherent junctions, which improve mTOR activity and guide bone colonization. VCAM-1 has been shown to bind osteoclast progenitor cells, inducing differentiation, and this cross-talk represents a critical step in the progression of microscopic bone metastasis to clinically significant bone metastasis [49]. However, there is no specific clinical application of the FOXO signaling pathway in existing studies.

In conclusion, by observing the phenotype of *wnt16* mutant zebrafish, we proved that the *wnt16* gene was expressed in zebrafish’s head, pectoral fin buds, and cartilage, and its deletion could lead to skeletal deformities and have a significant impact on bone mineral density and cortical bone. Furthermore, we also revealed complex changes in the expression patterns of all genes through RNA-Seq analysis, and proved that knocking out the *wnt16* gene would inhibit the expression of its associated signaling pathways FoxO and mTOR, and jointly regulate the Wnt signaling pathway, thereby affecting the ability of bone formation. It is found in our research that the *wnt16* gene affects bone development and bone formation, which provides more insights for the study of the *wnt16* gene mechanism. It may become a potential target for the treatment of osteoporosis.

## 4. Materials and Methods

### 4.1. Zebrafish Maintenance

All experiments were conducted using the AB wild-type strain of zebrafish (*Danio rerio*) of either sex. The animal protocols used in this investigation were approved by the Hunan Normal University Institutional Animal Care and Use Committee. Zebrafish larvae and adults were maintained at 28.5 °C in a recirculating aquatic system at a photoperiod of 14/10 h light/dark cycle [50].

### 4.2. Microinjection and Generation of Wnt16 Mutant Zebrafish

According to the principle of CRISPR/Cas9 [27], gRNAs against the *wnt16* were designed using a CRISPR design tool. Two gRNAs target sites are located in exon 3 of *wnt16*, and the target site sequence is as follows: a: GACACAAGCCTGTTGGGCAGCGG, b: GGCCTCCTCACCACGGGTCGAGG. The specific gRNA was synthesised using linearized p42250 vector as a template. An RNA mixture (1.8 nL) composed of gRNA (30 ng/μL) and Cas9 mRNA (500 ng/μL) was microinjected into one-cell stage zebrafish embryos. Embryos were then incubated with sterile E3 medium and raised at 28.5 °C. After 48 h, 5–10 injected embryos were picked to extract the genome, and PCR was performed to identify CRISPR-induced mutations. We use Primer 5.0 software to design primers, and the primer sequence was synthesized in Shanghai Shenggong Biological Engineering Co., Ltd. (Shanghai, China). PCR primers are AGAAATCTCTCAGCCAAACACG (forward primer) and ATACTGCGAACAATTCCTTGCT (reverse primer). The F0 generation zebrafish was obtained, and the F0–F1 generation zebrafish embryos and adult fish were then genotyped by the T7E1 endonuclease assay and DNA sequencing to determine whether mutations in *wnt16*-knockout progeny [28].

### 4.3. Whole-Mount In Situ Hybridisation

Whole-mount in situ hybridization was performed as previously described [51]. Embryos were treated with 1-phenyl-2-thiourea (PTU, Sigma, St. Louis, MO, USA) to prevent pigmentation until 3 dpf. The treated embryos in the corresponding period were fixed with 4% paraformaldehyde overnight. Total RNA was extracted from WT embryos at 24 h post-fertilization (hpf) and then reverse-transcribed to cDNA (Thermo, Waltham, MA, USA). The forward primer is 5′-AAAGAGACAGCGTTCATCCATG-3′, and the reverse primer is 5′-CATAACAGCACCAGACGAACTT-3′. DIG RNA labeling kit (Roche, Basel, Switzerland) and T7 RNA polymerase kit (Thermo) were used to synthesize RNA probes. The probe at a concentration of 5 ng/μL was added to a ribonuclease-free test tube and hybridized overnight at 65 °C. The next day, the larvae were washed and cultured with digoxin antibody at a dilution of 1:3000. On the third day, nitro blue tetrazolium/5-bromo-4-chloro-3 indole phosphate (NBT/BCIP, Roche) was used to mediate the color reaction. Three independent replicates were performed using at least 30 embryos each time.

### 4.4. Light Microscopy and Micro-CT Imaging

MESAB, Tricane, ddH2O, and Tris were used to anesthetize adult wild-type zebrafish (control) and mutant zebrafish with developmental abnormity of the same period until their sensitivity to activity was significantly reduced at pH 7.0. Then, they were observed and photographed under a stereoscopic microscope (SZX16). Subsequently, 4% of the paraformaldehyde was fixed in a 2 mL EP tube for 24 h, and then the zebrafish bones were analyzed by micro-CT using a Hamamatsu (L9181-02) microfocal X-ray tube. The ray tube voltage is 50 kV, the current is 300 μA, the scan angle is rotated 360°, the interval is 0.8°, a projection is collected, and the exposure time is 2 min [52]. After scanning, image data were transferred to a workstation, and vertebral bone mass and morphology demonstrated were calculated and viewed using 3Dmed software. The 3D models, which were from serial tomographic datasets, were used for visualisation and morphometric analysis of cancellous bone [53].

### 4.5. Library Construction and High-Throughput Sequencing

The total RNA was isolated from yolk-cell, 1-cell, and 4-cell embryos (three replicates each group) at WT and *wnt16^−/−^* zebrafish groups by the TRIzol method. The purity and concentration of total RNA were checked using Qubit (Life Technologies, Carlsbad, CA, USA) and Nanodrop (Thermo, Waltham, MA, USA), respectively, and RNA integrity was detected with Agilent 2100. The RNA-Seq sequencing library was prepared using TruSeq™ RNA sample preparation kit (lllumina, San Diego, CA, USA). Briefly, 2 mg of total RNA was used to isolate mRNA with oligo (dT) beads, and double-stranded cDNA was synthesized and purified using AMPure XP beads. After generating the clusters, library sequencing was performed on the Illumina HiSeq X-ten platform to create paired-end reads with a length of 150 bp (Novogene Biotech, Tianjing, China) [54].

### 4.6. KEGG and GO Enrichment Analysis of DEGs

The threshold of DEGs was set as *p* < 0.05 and |log2 (fold change)| > 1. To visualize the differential expression profiles of the enrolled RNAs, we used the “ggpubr” and “ggthemes” packages in R software to draw volcano maps. OmicShare (http://www.omicshare.com/tools/, accessed on 25 October 2020), a free online platform for data analysis, was used to perform hierarchical clustering analysis. The Venn diagram was used to screen out the differentially co-expressed genes by an online software (http://bioinformatics.psb.ugent.be/webtools/Venn/, accessed on 25 October 2020). To gain insights into the biological roles of these DEGs, GO and KEGG were used to analyze these up-regulated genes and down-regulated genes through the Database for the Annotation, Visualization, and Integrated Discovery (DAVID, Version 6.8, https://david.ncifcrf.gov/, accessed on 25 October 2020) [55,56]. Its cutoff value was *p* < 0.05. The “ggplot2” package in R software was used to visualize GO and KEGG enrichment results. The STRING database (https://string-db.org/, accessed on 25 October 2020) was used to infer protein–protein interactions (PPIs) in DEGs’ enrichment pathways [57].

### 4.7. Validation of Selected Genes by Quantitative RT-PCR

Eight down-regulated genes and *wnt16* were selected and detected by quantitative RT-PCR. At 48 hpf, 30 larvae were collected from the WT and *wnt16*^−/−^ groups, and total RNA was isolated using TRIzol reagent (Thermo Fisher Scientific) according to the manufacturer’s protocol. qRT-PCR was performed in Quantagene q225 system (Kubo, Beijing, China) using ChamQ Universal SYBR qPCR Master Mix kit (Vazyme Code: Q711-02). The thermal cycle is as follows: pre-denaturation at 95 °C for 30 s, then 40 cycles at 95 °C for 5 s, 60 °C for 15 s, and 60 °C extension for 30 s. The relative gene expression was normalized to the endogenous housekeeping gene (β-actin) and the formula 2^−^^△△Ct^ was calculated using the comparative Ct method (three biological replicates in each sample) [58]. The primer sequences used in this study are listed in Table 2.

### 4.8. Statistical Analysis

Statistical analysis was performed with GraphPad Prism software (version 8.4.3, GraphPad Software, Inc., La Jolla, CA, USA). All values were presented as the means ± SEM. Student’s *t*-test was performed to evaluate the statistical significance between two independent groups. Statistical significance was defined as a *p*-value less than 0.05.

## Figures and Tables

**Figure 1 ijms-22-06673-f001:**
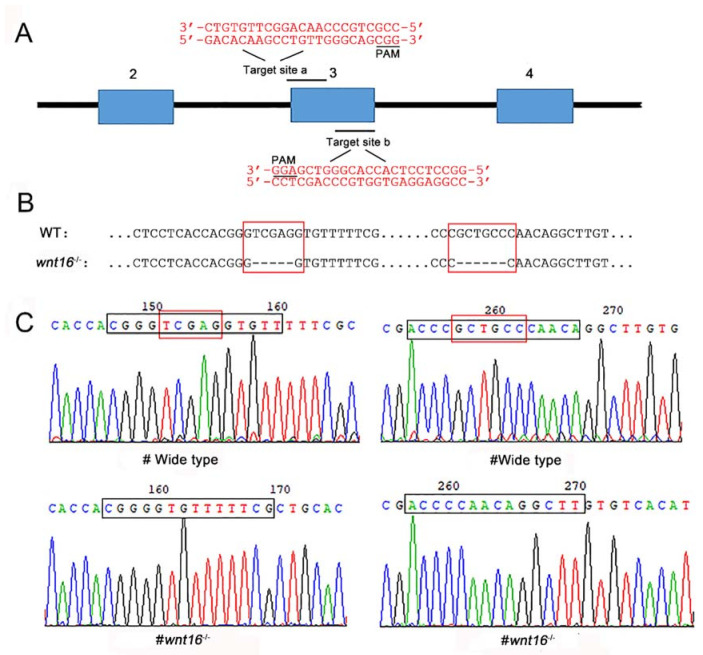
Generation of *wnt16* mutant zebrafish with the CRISPR/Cas9 system. (**A**) Schematic diagram of the target site in the zebrafish *wnt16* genome. (**B**) Sequence alignment between wildtype (WT) and *wnt16^−/−^* mutant. (**C**) Sequencing maps of WT and *wnt16^−/−^* zebrafish. Black frames: sequences of the target site. Red frames: TCGAG and GCTGCC in WT. Note the 11 bp (TCGAG and GCTGCC) deletion in *wnt16^−/−^*.

**Figure 2 ijms-22-06673-f002:**
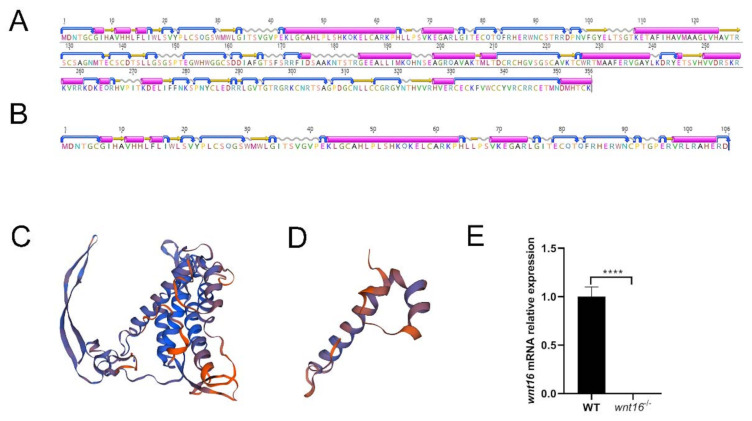
Structure prediction of *wnt16* protein and qRT-PCR verification. (**A**,**B**) Secondary structure prediction in WT and *wnt16^−/−^* zebrafish (purple cylinder indicates alpha helix, yellow arrow indicates beta strand, silver wavy line indicates coil, and blue turning arrow indicates turn). (**C**,**D**) Tertiary structure prediction in WT and *wnt16^−/−^* zebrafish. (**E**) qRT-PCR analysis of 2dpf WT and *wnt16^−/−^* larvae, showing a statistically significant decrease of *wnt16* expression in *wnt16^−/−^* zebrafish. The results are represented as means  ±  SEM, **** *p* < 0.001.

**Figure 3 ijms-22-06673-f003:**
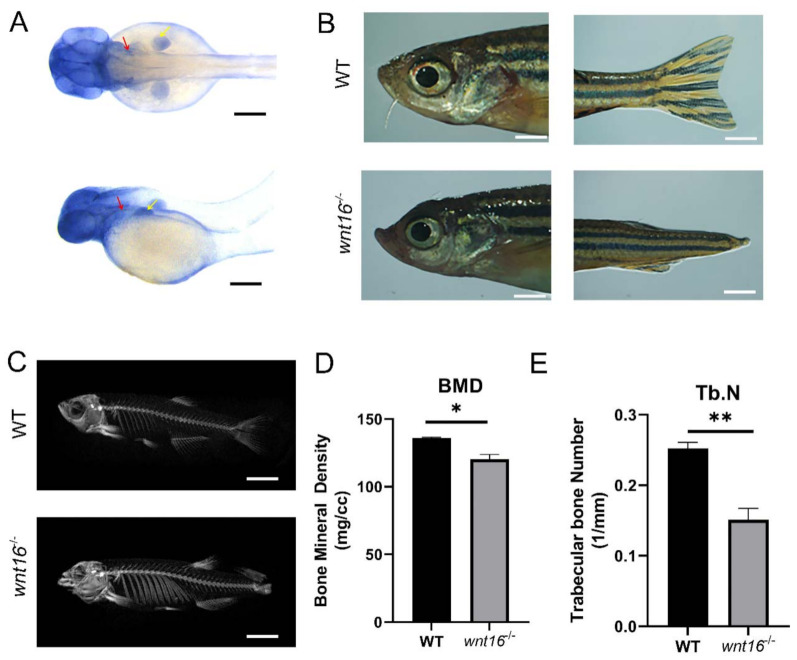
Images of zebrafish generated by WISH, light microscopy, and micro-CT and skeletal statistical analysis. (**A**) Images of whole-mount in situ hybridisation with *wnt16* mRNA probes in WT larvae at 48 hpf. Colour intensity is proportional to the expression level of *wnt16* gene. Arrowheads indicate relevant domains of expression. Yellow, pectoral fin bud (pfb); red, operculum (op). (**B**) Representatives of adult WT and *wnt16^−/−^* mutant zebrafish are shown. Apparent abnormality was observed in the head and tail of *wnt16^−/−^* mutant zebrafish. (**C**) Micro-CT scanning of WT and *wnt16^−/−^* mutant zebrafish skeleton. Obvious abnormality was observed in the mandible and spine of *wnt16^−/−^* mutant zebrafish. (**D**,**E**) Statistical analysis of BMD (**D**) and Tb.N (**E**) in WT and *wnt16^−/−^* zebrafish. Note that BMD and Tb.N value decreased in the *wnt16^−/−^* group. The results are represented as means  ±  SEM, * *p* < 0.05, ** *p* < 0.01. Scale bar: 150 μm (**A**); 2 mm (**B**); 6 mm (**C**).

**Figure 4 ijms-22-06673-f004:**
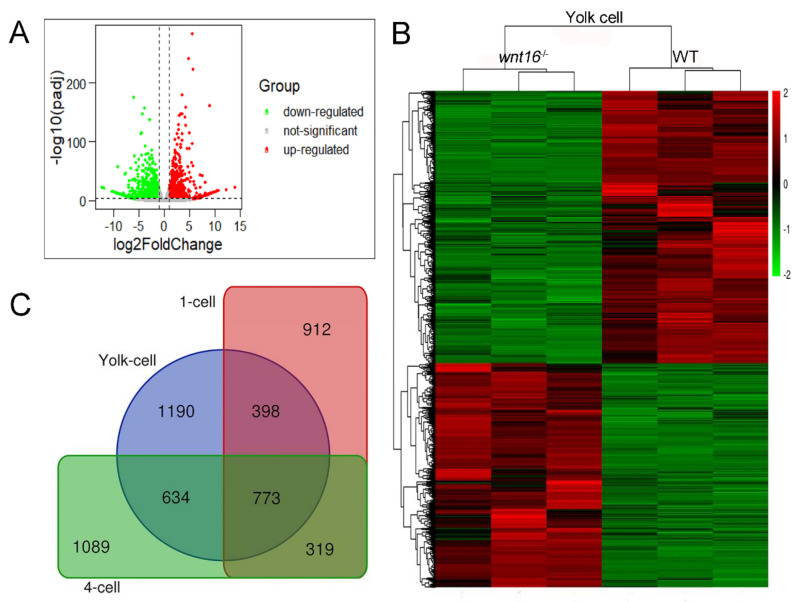
Clustering of differentially expressed genes (DEGs). (**A**) The volcano graph was performed to show DEGs in the yolk-cell period. The red part indicates up-regulated genes and the green part indicates down-regulated genes. (**B**) The overall distribution of DEGs between the WT group and *wnt16^−/−^* group in the yolk-cell period. Red and green represent up-regulated and down-regulated changes, respectively, in the clustering analysis. The color intensity is directly proportional to the change. (**C**) Venn diagram of the co-expressed differential genes of three groups.

**Figure 5 ijms-22-06673-f005:**
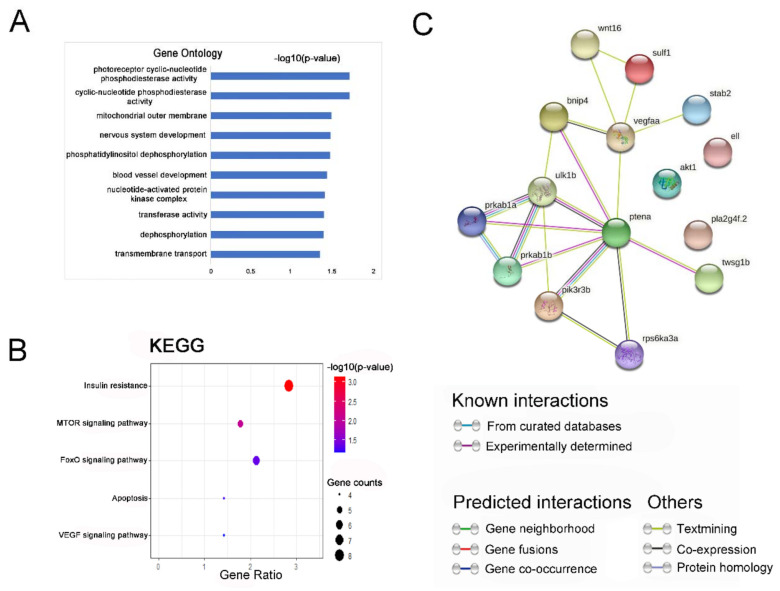
Gene ontology (GO) enrichment and Kyoto Encyclopedia of Genes and Genomes (KEGG) signaling pathway analysis of co-expression down-regulated DEGs. (**A**) GO analysis of co-expression down-regulated genes; bar plot shows the top ten enrichment score (−log10 (*p*-value)) of DEGs involving biological process, cellular component, and molecular function. (**B**) The significant changes in the KEGG pathway of co-expressed down-regulated genes. The bubble graph shows the enrichment score (−log10 (*p*-value)) of the significant pathway. The size of the circle represents the number of enriched DEGs. *p*-value was represented by a color scale, and the statistical significance increased from blue (relatively lower significance) to red (relatively higher significance). (**C**) Protein–protein interaction network of these fifteen DEGs and *wnt16*. Nodes represent genes, lines represent the interaction of proteins with genes, and the results within the nodes represent the structure of proteins. Line color represents evidence of the interaction between the proteins.

**Figure 6 ijms-22-06673-f006:**
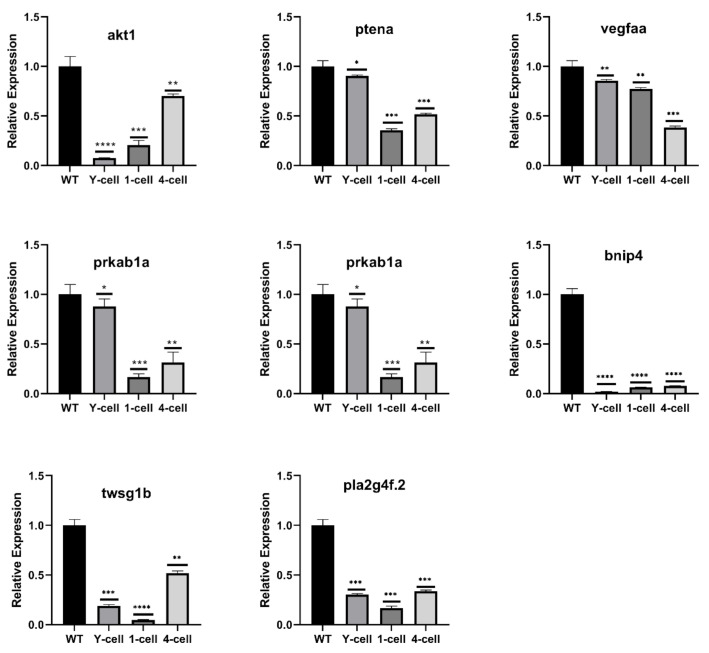
Validation of RNA-Seq data by qRT-PCR. Eight significantly down-regulated DEGs highlighted in GO enrichment and KEGG signaling pathway related to skeletal development. All the selected DEGs showed the same expression pattern in both RNA-Seq and qRT-PCR analysis. The results are represented as means ±  SEM, * *p* < 0.05, ** *p* < 0.01, *** *p* < 0.002, **** *p* < 0.001.

**Table 1 ijms-22-06673-t001:** Down-regulated genes gathered in gene ontology (GO) enrichment and Kyoto Encyclopedia of Genes and Genomes (KEGG) signaling pathway related to skeletal development.

Signaling Pathway	Genes
GO:	
Blood vessel development	*ptena*, *twsg1b*, *vegfaa*, *stab2*
Vasculogenesis	*ell*, *sulf1*, *vegfaa*
KEGG:	
mTOR signaling pathway	*pik3r3b*, *akt1*, *rps6ka3a*, *ptena*, *ulk1b*
FoxO signaling pathway	*pik3r3b*, *prkab1a*, *akt1*, *prkab1b*, *ptena*, *bnip4*
VEGF signaling pathway	*pik3r3b*, *akt1*, *vegfaa*, *pla2g4f.2*

**Table 2 ijms-22-06673-t002:** The primer sequences of the down-regulated genes and *wnt16* of qRT-PCR.

Gene	Forward Primer (5′-3′)	Reverse Primer (5′-3′)
*wnt16*	TCCTCACCACGGGTCGAG	CACCGAGGGCTGGCATTG
*β-actin*	ACGAACGACCAACCTAAACTCT	TTAGACAACTACCTCCCTTTGC
*akt1*	GGTCCTGATGATGCGAAAGA	CTTGAACGGAGGAACCAACT
*ptena*	GTTGCCCTCCTCTTCCATAAA	GGATTCACCTCACTCCTGTTT
*vegfaa*	TGTAAAGGCTGCCCACATAC	TGCTCGATCTCATCGGGATA
*prkab1a*	CAGTCCCGAAGATGCTGATATT	AACACAGTGGGTCGATCTAAAG
*prkab1b*	GACAAGATCAGGAGTCGGATAG	GAAGGAGCCAGACAAGTAGAT
*bnip4*	CTCGTGGGTAGAGCTTCATTT	CTGCTACAGTGAGAGCTTGTT
*twsg1b*	CTCGTGCTGTAAGGAGTGTATG	AACAGATCCTCCACCGTACT
*pla2g4f.2*	GTAACCTCACACCTGGACAAA	AGCTCACAGTCATCTCAAACTC

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
