# Peer review of "Loss of Wnt16 Leads to Skeletal Deformities and Downregulation of Bone Developmental Pathway in Zebrafish"

_ijms, 2021, doi:10.3390/ijms22136673_

Round 1

Reviewer 1 Report

This research manuscript entitled as “Loss of Wnt16 Leads to Skeletal Deformities and Downregulation of Bone Development Pathway in Zebrafish” offers a novel approach to discover the critical role of Wnt signaling in bone development of Zebrafish.  The article is a great example of the effective use of CRSPER-Cas9 technology in the field of developmental biology. The concept of the research article is in alignment with the aim and scope of the journal and hence I think it is appropriate to consider this review for the publication upon minor improvement in the current format of the manuscript.

The work reviewed in this review is interesting for broad audience and it definitely highlights the significant progress made in the developmental biology of Zebrafish. The article brings up some insightful findings which offers attractive ideas for researchers working on development of tools to investigate the gene signaling pathways in physiology and pathology of bone development. The methodology used for acquiring data from various molecular and imaging techniques and gene repositories is appropriate and authors have attempted to sufficient literature to cover the critical information. 

Authors have made sure to maintain the flow of information systematically making it appealing and intriguing for wide range of audience. The figures presented in the research offer a comprehensive summary of the big data analysis but needs to be upgraded for high-quality experience.

Overall, authors have done commendable work to gather a huge amount of information in compiling this research, elegant presentation and addressing the critical issues can improve this review for publication quality. I recommend following suggestions to be considered in order to revise this manuscript to be considered for publication in this journal.

Specific Comments:

  • In the introduction, authors propose to investigate the molecular mechanism of the wnt16 as a potential target for the treatment of osteoporosis but did not mention any background of current or ongoing therapeutic modalities that target wnt16 signaling in treatment of osteoporosis. I think such background is necessary given the therapeutic importance of the investigation undertaken in this study.
  • In the methods, 4.2 section. It is not clear- how the primers were obtained and synthesized. Please provide the source of each primer, enzymes and diagnostic kits used in this study.
  • In the methods, 4.4 the method to anesthetized the zebrafish is poorly written. Authors need to provide the details of anesthesia procedure- i.e. name of the anesthetic used, concentration, route of application and duration.
  • Except figure 1 and 6, other figures have resolution issues and the labels are very small and difficult to read after printing the paper.
  • While the data is interesting methodologically but what is missing is the relevance to human disease biology of wnt16 gene. Especially in the discussion, authors need to provide references citing the significance of their findings in clinical context along with the outlook of how this finding could potentially be a new drug target.

Reviewer 2 Report

The authors write: Previous studies have shown that the role of FoxO in chondrocytes is essential for the normal de-velopment of bone, and the role of FoxO in osteoclasts or osteoblast greatly affects bone resorption and formation, thereby affecting bone mass[32, 33]

I recommend to give information about the origin of bone mass (rat, chicken ?).
